# Influence of Country Digitization Level on Digital Pandemic Stress

**DOI:** 10.3390/bs12070203

**Published:** 2022-06-22

**Authors:** Álvaro Antón-Sancho, Diego Vergara, Pablo Fernández-Arias

**Affiliations:** 1Department of Mathematics and Experimental Science, Catholic University of Ávila. C/Canteros s/n, 05005 Ávila, Spain; alvaro.anton@ucavila.es; 2Department of Mechanical Engineering, Catholic University of Ávila. C/Canteros s/n, 05005 Ávila, Spain; pablo.fernandezarias@ucavila.es

**Keywords:** digital competence, stress, adaptation skills, COVID-19, digital skill

## Abstract

A quick and effective solution to address the immediate change in teaching methods after the COVID-19 pandemic was digital learning environments (DLEs). The way in which this process of change towards DLEs was tackled was different around the world, depending on multiple factors, including the level of digitization, technology, and innovation. This paper conducts quantitative research on the influence of the COVID-19 pandemic on the adaptation of university professors to DLEs. In order to achieve this objective, a sample of 723 university professors from 15 countries in Latin America and the Caribbean was taken. The participants’ self-perception of the stress generated and their levels of digital competence during the COVID-19 pandemic were studied according to the Global Innovation Index (GII) of their country of origin. The results show that professors have an intermediate–high self-perception of both their digital competence and their ability to adapt to DLEs. It is also shown that neither the professors’ level of digital competence nor the GII of the country of origin fully explain the level of pandemic stress regarding the use of DLEs. This fact suggests that there must be other influential factors to consider, thus opening new lines of future research.

## 1. Introduction

On 11 March 2020, the World Health Organization (WHO) declared the outbreak of COVID-19 a pandemic [1]. The confinement suffered in most countries around the world during the year 2020 because of the pandemic generated by the SARS-CoV-2 virus had an immediate impact on today’s globalized society, causing rapid changes in their different ways of relating to each other, as well as in their professional activities [2,3,4]. One of the professional groups that suffered most intensely from these changes was teachers who carried out their teaching activities at different educational levels. The impossibility of continuing to teach face-to-face classes generated an abrupt transformation of the educational system in a matter of hours [5]. More than 1.5 billion students worldwide were affected by the closure of schools or universities during the initial phase of the COVID-19 pandemic in 2020, and distance learning was introduced for many students [6]. Almost immediately, more than 90% of the global student population and millions of teachers worldwide adapted to online teaching [7,8], even in training activities where this digital migration is particularly complex, due to the technical circumstances involved, such as, for example, laboratory practicals [9,10].

In order to carry out this rapid adaptation of the educational system towards virtuality, it was necessary to invest considerable economic and technical resources and to rapidly acquire numerous electronic devices and peripherals, i.e., webcams, headsets, and virtual communication platforms. The great development of Information and Communication Technologies (ICT) during the years prior to the pandemic was key to coping with the confinement situation [11]. A quick and effective solution to cope with the immediate change of the didactic approach after the COVID-19 pandemic was digital learning environments (DLEs). DLEs are both a system of new tools and methods for teaching activity and a prospective educational ecosystem for learning [12]. Significant infrastructure for online education existed in many countries prior to the pandemic; however, long before the COVID-19 pandemic, parts of the scientific community noted discrepancies between the promises and improvements brought by educational technology [13]. In general, universities were not prepared for such an abrupt shift to online education as the pandemic did. Prior to COVID-19, many educational activities were conducted online, such as student registration, class scheduling, course scheduling, taking tests and exams, etc. However, with the advent of COVID-19, some of the learning platforms used prior to the pandemic, such as DingTalk, Tencent Meeting, ZoomCloud, TIM, WeChat Work, Chaoxing Learning, and MOOCs, become post-pandemic as complete course and user management tools [14] and online communication enablers [15].

This transformation highlighted the weaknesses of the educational system, especially the lack of virtual learning resources since the system was mainly based on the presence of all the actors involved in the teaching-learning process [16]. This migration even took place in the context of the temporary closure of many schools and universities [17,18]. However, it also revealed the inequality between countries. While the more developed countries were able to adapt quickly to the circumstances, other less developed countries had serious problems continuing teaching [19]. Millions of teachers had to face the abrupt virtualization of the educational system, sometimes with a shortage of resources, both ICT and the right jobs [20]. This situation was seen by the teachers themselves as an assessment of their professionalism [21], while the way of dealing with this process of change towards DLEs in the different countries of the world was influenced by multiple factors, among which the level of digitalization, technological development and innovation stood out.

The Global Innovation Index (GII), published by Cornell University, the Institut Européen d’Administration des Affaires (INSEAD), dedicated to business administration, and the World Intellectual Property Organization (WIPO), analyzes 131 countries around the world from different points of view that define the innovative character of their respective economic developments [22]. Each country is assigned an index in a range from 0 to 100 that centralizes many dimensions affecting innovation. Among these aspects, technological and digital knowledge and development are included, and technical and digital aspects linked to innovation are considered as one of its main aspects, so the GII is often taken as an indicator of the innovation and digitalization of countries [23,24,25]. 

The GII distinguishes seven different geographical zones, within each of which countries are expected to have similar economic developments, which justifies comparing digitization levels through the GII: Europe; Northern Africa and Western Asia; South East Asia, East Asia, and Oceania; Northern Africa and Western Asia; Sub-Saharan Africa; Northern America; and Latin America and the Caribbean [22]. The latest published GII, which corresponds to the year 2021, yields the results shown in Figure 1 for the area of Latin America and the Caribbean [22]. The mean GII for the area is 29.29, and the standard deviation is 4.55. Based on these statistics, three areas have been defined (hereafter referred to as GII areas) that are distinguished by their GII within the Latin America and the Caribbean area:Low GII: Corresponds to countries with a GII smaller than the mean GII minus the standard deviation (24.73); it consists of Guatemala, Bolivia, and Honduras;Intermediate GII: Countries with GII between the mean GII minus the standard deviation (24.73) and the mean GII plus the standard deviation (33.84); it consists of Uruguay, Colombia, Peru, Argentina, Panama, Paraguay, Ecuador, and El Salvador;High GII: Countries whose GII is greater than the mean GII plus the standard deviation (33.84); it consists of Chile, Mexico, Costa Rica, and Brazil.

Different studies carried out in recent months have analyzed professors’ perceptions of their own self-efficacy [26] and their level of digital competence [27,28,29], identifying the need to implement professional development to develop an optimal level of digital competence in the teaching staff [30], as well as the effects that the use of ICTs has on professors [31] and the future challenges implied by their implementation [32]. For its part, the GII has been studied as a discriminating variable of the perceptions of certain families of skills of university professors, among which is digital competence [33]. On the other hand, having a low level of digital competence leads to an increase in perceived stress, as well as a reduction in professors’ motivation [34].

The analysis of the emotional effects derived from the digital impact caused by the pandemic is a topic that the specialized scientific literature deals with intensively. Indeed, numerous very recent studies have concluded that there is a psychological impact caused by the necessary digitization of learning environments resulting from the pandemic among university professors [31,34]. In addition, personal factors affecting the intensity of this impact have been identified [33]. Among higher education students, previous studies conclude that students are prepared to withstand the digitalization process they have experienced due to the effort in incorporating digital technologies that different countries have been making in recent years [35,36]. However, it is detected that the emotional impact (increased anxiety, tension, or worry) has increased and has done so in different ways according to different circumstances, including the level of innovation and policies of the country [37].

The preceding literature is also concerned, albeit less frequently, with analyzing the affective impact of pandemic-induced digitization among university professors. Typically, these analyses focus on well-defined countries or geographic areas and highlight the concern of university professors about the negative formative consequences of forgoing face-to-face training [38]. Regarding the psychological impact, faculty generally report that the pandemic has caused a major disruption in the functionality of their academic activities, which has had a major impact on their anxiety and personal lives [18]. However, studies indicate that there are factors related to the personality of teachers that condition the psychological impact suffered during the pandemic due to the digital migration of their activities [39]. In any case, the scarce digital training that university professors classically show can be identified as a common factor that explains the psychological impact suffered [40]. In Latin America, which is the area studied in this article, it has been found that there is a positive relationship between digital competence and stress during the pandemic among teachers (i.e., the highest level of stress is found among those with greater digital competence) [41]. In addition, a strong territorial gap in the level of digitalization has been identified, although no studies have been found that analyze the influence of this territorial gap on the psychological impact under analysis [42,43].

In view of this situation, the main goals were to analyze the psychological impact of Latin American and Caribbean university professors toward DLEs during the COVID-19 pandemic and analyze the existence of possible statistically significant gaps in these perceptions due to the level of digitization of the participants’ country of origin. In addition, as secondary variables, we will study the existence of differences by gender, age, and area of knowledge in the above perceptions. Furthermore, taking into account that the population analyzed in this study is related to one of the professional sectors that have had to make the most intense digitization efforts after COVID-19, the obtained results could be extrapolated to other professional sectors that have also undergone the digitization effort, i.e., the results of the study are generalizable to more sectors than just the education sector. In order to do this, the results obtained on different self-assessments of the professors are analyzed: (i) assessment of own digital competence; (ii) perception of the professional aspects; (iii) perception of the level of stress; (iv) self-confidence in the professor’s work; and (v) self-concept on adaptation skills to DLEs, according to the GII. In this analysis, it is assumed that the assessments are multivariate normally distributed within each group of the considered characteristics of the sample—GII level, gender, age, and area of knowledge—and that the population covariance matrices of each group are equal. 

The work is structured in four main sections. The Materials and Methods section describes the sample of participants in the study, the instrument used, the research objectives, variables, and hypotheses, and explains the research methodology. The Results section presents the results obtained from the statistical analysis of the responses to the survey employed. The Discussion describes the main results obtained and relates them to each other and to the previous literature, indicating limitations and future lines of research. Finally, the main conclusions of the work are presented in the last section of the Conclusions.

## 2. Materials and Methods

### 2.1. Participants

A total of 723 university professors who work in the 15 countries of Latin America and the Caribbean that have GII participated in this study: Chile, Mexico, Costa Rica, Brazil, Uruguay, Colombia, Peru, Argentina, Panama, Paraguay, Ecuador, El Salvador, Guatemala, Bolivia, and Honduras. These participants were selected through a non-probabilistic convenience sampling process. The professors were contacted by e-mail and were sent the survey that was used as a research instrument through GoogleForms^TM^. Participants responded to the survey voluntarily, freely, and anonymously. All responses were validated (a response is considered valid when it is complete).

### 2.2. Objectives, Variables and Hypotheses

The general objective of this research is to analyze the influence that the migration to university teaching DLEs caused by the COVID-19 pandemic has had on the self-concept of professors from different GII areas in Latin America and the Caribbean regarding their digital skills and the confidence they have in their own abilities to adapt to these digital environments. In particular, the following specific objectives are pursued: (i) to analyze the perception that Latin American and Caribbean university professors have about their own digital skills, self-confidence, and ability to adapt to DLEs and about the stress caused by the need to increase the use of digital environments due to the COVID-19 pandemic; (ii) to study whether there are significant differences between professors from the different GII areas in the above respects; and (iii) to identify significant gaps by gender, age or area of knowledge in the analyzed perceptions among professors from each GII area. In the present study, the following independent variables are considered in relation to the sample of participants: (i) GII area to where the country in which he/she teaches belongs; (ii) gender; (iii) age; and (iv) area of knowledge. All the variables mentioned are nominal. The first variable (GII area) is trichotomous and can take the values High GII, Intermediate GII, and Low GII, according to the classification of GII areas that have been defined. The gender variable is dichotomous (with values female and male). Age is defined as a polytomous variable whose values correspond to the following age ranges: 25 to 34 years old, 35 to 44 years old, 45 to 54 years old, and 55 years old or older. Finally, the following areas of knowledge are distinguished: Arts and Humanities (covering the fields of art, philology, philosophy, and history; hereafter, Humanities); Science (mathematics, physics, chemistry, and natural sciences); Health Sciences (fields related to medicine; hereafter, Health); Social and Legal Sciences (geography, economics, sociology, law and education; hereafter, Social Science); Engineering and Architecture (referring to technical education; hereafter, Engineering). The areas of knowledge have been defined according to the International Standard Classification of Education (ISCED) of the United Nations Educational, Scientific and Cultural Organization (UNESCO) [44], but integrating Education within the area of Social and Legal Sciences and distinguishing an area for Health Sciences.

The dependent variables that have been considered in this study are as follows:Assessment of own digital competence for the use of DLE use during the pandemic.Perception of the professional aspects linked to the use of DLEs during the pandemic (help from the university, professors and students, adequacy of spaces, and technical and human resources).Perception of the level of stress caused by the pandemic with respect to the professional practice of teaching using DLEs (hereafter referred to as pandemic digital stress).Self-confidence in the professor’s work during the pandemic.Self-concept on adaptation skills to DLE use during the pandemic.

All dependent variables are ordinal quantitative and have been measured on a scale from 1 (lowest valuation) to 5 (highest valuation). Throughout the study, the following hypotheses are demonstrated: (i) within the Latin American and Caribbean areas, the level of digitalization of the country significantly influences the self-concept of digital competence and digital pandemic stress expressed by the participants; and (ii) there are gender, age, and area of knowledge gaps in the perceptions of digital pandemic stress suffered.

### 2.3. Instrument

In order to achieve the objectives of this study, a survey of 38 questions or items was used. The survey was designed for this purpose and based on the instrument developed in [29] for the measurement of digital pandemic stress on professors. This survey was employed to measure the values of the different dependent variables defined as indicated in Table 1. All the questions are Likert-type from 1 to 5, where 1 means the lowest valuation of the aspect whose assessment is requested and 5 means the highest valuation.

### 2.4. Statistical Analyses

This paper is quantitative descriptive research based on the analysis of the answers to a survey of 38 Likert-type questions valued from 1 to 5, which has been used as a research instrument. The validation of this instrument was carried out by means of an Exploratory Factor Analysis, which resulted in the definition of five subscales that explain the global survey. The psychometric validation of the instrument was obtained by computing the Pearson correlation coefficients of the different subscales among themselves and with respect to the global survey. Additionally, the convergent validation was analyzed by means of the average variance extracted (AVE) and the internal consistency of the survey through the Cronbach’s alpha parameters and composite reliability (CR) of the subscales. The responses were analyzed using descriptive statistics (mean and standard deviation). Participants were differentiated by the GII area to which their university belongs, and the means of the different subscales of the survey were compared using the ANOVA test and the standard deviations using Levene’s test. Finally, the participants were differentiated by GII area and by the rest of the independent variables of the study (gender, age, and area of knowledge), and the mean responses of the different subscales were compared using the multifactor ANOVA test (MANOVA). When it was not possible to assume homoscedasticity, the mean comparison tests were applied with Welch’s correction without assuming equal standard deviations. All tests were performed with a significance level of 0.05.

## 3. Results

### 3.1. Main Characteristics of the Total Sample

As shown in Figure 2, there is a majority of participants from countries with high GII, while the smallest proportion is from countries with low GII. Figure 3 shows the distributions of the sample of participants when differentiated by the GII area and the rest of the independent variables. Females are in the majority in countries with low or high GII, while in countries with intermediate GII, the majority are women, the differences in proportions being statistically significant (chi-square = 9.5027; *p*-value = 0.0086). The most frequent age range in countries with high GII is between 35 and 44 years, while in countries with intermediate GII, it is between 45 and 54 years, and in those with low GII, the two central age ranges have the same frequency (chi-square = 16.405; *p*-value = 0.0117). Finally, in countries with high GII, the areas of knowledge with a majority representation are Humanities and Social Sciences, while in countries with intermediate or low GII, there are more professors of Social Sciences and Engineering (chi-square = 119.53; *p*-value < 0.0001).

### 3.2. Validation of the Instrument

The EFA, performed with Varimax rotation on the survey responses, identifies five factors that explain the survey (Table 2). The factors identified correspond to the dependent variables of the study. Hereafter, each of these factors will be referred to as subscales of the overall survey. The above definition of subscales explains 61.1% of the total variance (Table 3). From the Cronbach’s alpha parameters and CR shown in Table 4, it is deduced that the internal consistency of the subscales is high. The convergent validity analysis was performed by means of the AVE values, which are adequate.

For the psychometric validation of the instrument, the Pearson correlation coefficients of the different subscales defined in the EFA model were computed. As can be noted in Table 5, the subscales of the survey correlate weakly (very weakly in the case of the digital pandemic stress subscale with respect to the rest of the subscales). On the other hand, there are intermediate or high correlations between the different subscales and the global survey.

### 3.3. Analysis of Responses

In general, participants express intermediate–high levels of digital competence and ability to adapt to DLEs (Figure 4). These subscales also have the lowest deviations, indicating that participants are more confident in their responses than in the rest of the subscales. They also give intermediate valuations of the professional dimensions related to migration to DLEs and of their own levels of self-confidence, although, in these subscales, the valuations are slightly lower than in the digital competence and adaptation skills subscales. Finally, the assessment of digital stress caused by the pandemic is intermediate–low, even though this subscale is the one with the highest standard deviation and, consequently, the greatest heterogeneity of responses.

By GII areas, Figure 5 shows that participants from countries with low GII are those who report having greater digital competence and ability to adapt to DLEs during the pandemic. The ANOVA test shows that these differences are significant for both the digital competencies subscale (F = 4.5949; *p*-value = 0.0102) and the adaptation skills subscale (F = 13.8420; *p*-value < 0.0001). However, participants from countries with high or intermediate GII value professional aspects linked to DLE use more highly (F = 11.0510; *p*-value < 0.0001) and express significantly higher levels of self-confidence than participants from countries with low GII (F = 3.5111; *p*-value = 0.0301). Regarding the perception of pandemic stress related to the use of DLEs, professors from countries with high GII expressed higher levels of stress, and those from countries with low GII expressed lower levels of pandemic stress. 

Levene’s test of standard deviation comparison shows that homoscedasticity can be assumed in the digital competence subscale when differentiated by the GII area (F = 0.2780; *p*-value = 0.7573). For the rest of the subscales, homoscedasticity cannot be assumed. Professors from countries with low GII are more confident than the rest in the assessment of professional aspects linked to DLEs (sd = 1.07 over 5, compared to sd = 1.14 from the area with high GII and sd = 1.15 from the area with intermediate GII, with Levene’s statistic F = 3.8187; *p*-value = 0.0220). Regarding the level of adaptation to DLEs, self-confidence, and the level of pandemic stress linked to the use of DLEs, the greatest heterogeneity of responses is found in countries with high GII. The largest differences between deviations are in the adaptation skills subscale (sd = 0.97 over 5 in the area with high GII, sd = 0.88 in the area with intermediate GII, sd = 0.90 in the area with low GII, with Levene statistics F = 5. 7665; *p*-value = 0.0032) and in the self-confidence subscale (sd = 1.08 over 5 in the area with high GII, sd = 1.05 in the area with intermediate GII, sd = 0.98 in the area with low GII, with Levene statistics F = 6.7397; *p*-value = 0.0012). This means that the widest gap between GII areas in terms of participants’ confidence in responding is in the subscales of adaptation to DLEs during the pandemic and self-confidence. The gap between GII areas in terms of deviations is smaller in the subscale measuring digital pandemic stress (sd = 1.21 over 5 in the area with high GII, sd = 1.18 in the area with intermediate GII, sd = 1.15 in the area with low GII, with Levene statistics F = 3.2305; *p*-value = 0.0396).

The MANOVA test shows that there are gender gaps in the responses to all subscales of the survey (Table 6). From the mean data shown in Table 6, the superiority observed in the low GII area in relation to digital competence is found among females. For males, the self-assessment of digital competence is higher in countries with high or intermediate GII (Figure 6). Likewise, the lower valuation of professional aspects linked to the use of DLEs observed in countries with low GII is found among males, while among females, those who value these aspects the least are those in countries with high GII. Females express a better capacity to adapt to DLEs during the pandemic than males. In fact, females from low GII countries express better adaptation than those from the rest of the GII areas, while, among males, those from low GII countries express worse adaptation. The highest levels of self-confidence are shown by professors from countries with intermediate GII among males and high GII among females. Finally, among males, professors from countries with low GII have suffered greater digital pandemic stress, while, among females, it is the female professors from countries with high GII who express higher levels of stress.

By age range, the lowest levels of digital competence are expressed by those over 55 years old, although professors in countries with low GII are more digitally competent than those in the rest of the areas in all age ranges (Table 7 and Figure 6). Professors in countries with intermediate GII value the professional aspects linked to the use of DLEs during the pandemic more highly than professors in countries with low GII, although the latter outperform those with high GII in their assessment in the 35 to 44 age ranges and among those over 55 years old. In contrast, professors from countries with high GII show less ability to adapt to DLEs during the pandemic than those from the rest of the areas in all age ranges (except among those under 35 years old, in which the average assessment coincides with those from countries with low GII). The lowest levels of self-confidence are found in professors from countries with low GII in the age range under 45 years old. Among those over 45 years, the self-confidence of professors in countries with low GII exceeds that of professors in the rest of the GII areas, except among those over 55 years old, among whom those in countries with low GII exceed those in high GII, although not those in intermediate GII. Finally, the greatest pandemic stress is found in countries with low GII, except among those under 35 years old, among whom the greatest pandemic stress is found in countries with high GII. The differences observed by age range are statistically significant in all subscales (Table 7).

The area of knowledge is another discriminative variable for all the subscales analyzed, as shown by the MANOVA test statistics (Table 8 and Figure 7). A significant gap is observed between professors in the humanistic-social areas (Humanities and Social Sciences) and those in the scientific-technical areas (Science, Health Sciences, and Engineering) with respect to the digital competence expressed by the professors. Among the former, those from countries with low GII express the highest digital competence, while among the latter, those from countries with low GII express the lowest GII. A similar phenomenon occurs with the self-confidence subscale, except in Health, where professors from countries with high GII give the lowest evaluation. As for professional aspects, professors from countries with high or intermediate GII give higher assessments than those from countries with low GII, except in Social Science. In contrast, teachers in countries with low GII report having adapted better than those in countries with high or intermediate GII to DLEs during the pandemic, except in Science and Engineering. Professors in countries with intermediate GII have experienced the least digital pandemic stress in the areas of Science, Humanities, and Health Sciences. Those with the greatest stress are those in countries with low GII in Science and Health and those with high GII in Humanities. In Social Science, professors from countries with high GII are those who show the greatest pandemic stress. In Engineering, there are hardly any differences in this regard, but professors from countries with high GII are those who express lower levels of stress due to the use of DLEs during the pandemic.

## 4. Discussion

### 4.1. Main Results and Relationship with Previous Literature

Although the sample of participants on which the research was conducted contains most professors from countries with high or intermediate GII (Figure 2), which is consistent with the fact that countries with low GII are less numerous and have smaller populations than many of the countries with high or intermediate GII [45,46]. By gender and age (Figure 3), the distribution of participants does not correspond to the distributions of the general population of Latin America and the Caribbean, where there is, in general, a slight majority of men and younger age groups [45]. This result is to be expected because the participants are part of a very specific population sector (university professors) whose demographic characteristics differ from those of the overall population. By areas of knowledge, it is notable that countries with low GII have an overrepresentation of Science professors and an underrepresentation of Engineering and Architecture professors with respect to the other GII groups (intermediate and high). However, it is not possible to say, in principle, whether these results are generalizable to all professors in the Latin American and Caribbean region (i) because, as far as we have been able to explore, there are no published data available on the distribution of professors by subject area in the region and (ii) because the process by which the participants were chosen is not probabilistic.

The specialized literature shows that the stress caused by the COVID-19 pandemic has a decisive influence on various aspects of the professional activity of university professors, such as research activity [47]. The results of this work support that observation when measuring stress levels referred to the adaptation of university professors to the use of DLEs. The average levels of expressed digital competence are intermediate–high. This fact suggests that it is necessary to improve the way in which university professors are being trained in digital competence. In addition, the expressed digital competence and the level of adaptation to digital environments are lower the higher the GII of the country of origin (Figure 4). The fact that professors report intermediate levels of digital competence is in line with other work performed on digital skills of professors in the same geographical area, both during the pandemic [48,49,50] and prior to it [51]. According to the Inter-American Development Bank report, university professors in Latin America and the Caribbean identify a lack of digital training and difficulty in accessing technological resources as the main difficulties in the use of digital environments in higher education learning processes [52]. Some published research coincides with the conclusions of the previous report in this regard [53]. The previous literature does not include results on differences in digital competence by region or country with different levels of innovation or digitization, so it is not possible to discuss the differences identified here by GII with respect to previous works. In any case, the fact that countries with high GII are those that express the lowest level of digital competence may be due precisely to the fact that the greater degree of digitization they enjoy leads to a more realistic view on the part of professors about the limitations of their digital skills. This may be the reason why, despite feeling less digitally competent, representatives of countries with high GII value the technical and professional aspects of DLE use more highly (Figure 4). This conclusion should be contrasted in a subsequent study to assess whether the perceptions expressed by teachers correspond to their objective digital competence or whether, on the contrary, there really is environmental conditioning that influences the self-concept of digital competence. On the other hand, the professors’ levels of digital competence are like those expressed by students in the Latin American and Caribbean areas [54]. 

In the context described above—greater pessimism about one’s own skills expressed by professors from countries with high GII—the levels of pandemic stress due to DLE use are intermediate but notably higher in countries with extreme levels of GII—low or high—than in countries with intermediate GII (Figure 4). Likewise, the personal level of digital competence has little influence on the stress derived from DLE use, as demonstrated by the weak correlations shown in Table 5. The above trends are expressed visually in Figure 5. Several conclusions can be drawn from the above: (i) there is a certain negative correlation between country GII and personal digital competence expressed by professors; (ii) there must be other sociological, economic, political, or cultural factors that affect the perception of digital competence beyond country GII; (iii) country GII has more influence on levels of DLE pandemic stress than self-concept of digital competence; and (iv) there is no direct correlation, either positive or negative, between GII and levels of DLE pandemic stress. Consequently, it seems that there are latent factors other than those considered in the GII, that should also contribute significantly to shaping levels of digital pandemic stress. 

Numerous studies have found that the migration to digital environments of teaching processes caused by the pandemic has increased the workload of teachers and has generated difficulties in adapting to online environments, which has generated levels of anxiety associated with teaching [29,55,56,57]. There are also studies that corroborate that pandemic stress has had an impact on the adaptation to digital environments of university students in different areas of knowledge [58,59]. The moderate levels of pandemic stress due to DLE use obtained in the present study are in line with those of previous works. They are also in line with the levels of digital anxiety that the preceding literature attributes to undergraduate students [37], although they report higher levels of digital competence than have been found among professors [35,36]. The intermediate valuations that the participants give to the professional aspects linked to the use of DLEs, which are lower than those they give to their digital competence (Figure 4), confirm that part of the stress expressed is due to the fact that they feel a certain dysfunctionality in academic activities caused by digital migration. This is in line with the preceding literature [18] and with a certain reluctance towards non-presential education, also supported by previous studies [38]. Finally, it has been shown that the level of digitalization of the country is one of the elements that explain the territorial gap that the literature had already found for Latin America and the Caribbean [42,43].

The above observations suggest the need to design digital skills training plans aimed at generating solid digital skills training for professors that can induce an increase in students’ digital skills [60]. These efforts in faculty training should be accompanied by an increase in investment in technical and human resources linked to the use of DLEs, given that in this variable the countries with low GII also have the lowest scores (Figure 4). These necessary acts of training in digital competence can be achieved by integrating the teaching and research aspects that make up the professional development of teachers. Indeed, there are many digital tools that can be used in both teaching and research, such as content sharing platforms, meeting tools, presentation software, or technical software specific to each area of knowledge. Training in digital competence that integrates the teaching and research dimensions in this sense will help professors to give a more unified character to their activity and to lower the levels of digital stress linked to both activities. These suggestions are in line with the recommendations of the Organization for Economic Cooperation and Development (OECS) regarding the development of digital skills among teachers in Latin America [61].

Differences were also identified by gender, age, and area of knowledge in the assessments of all the variables studied. By gender, women reported greater competence and a higher ability to adapt to DLEs than men (Figure 6). In addition, women express a similar or lower level of digital pandemic stress than men, except in countries with intermediate GII (Table 6). All this implies that the gender gap in the perceptions analyzed generally benefits female professors in terms of their training and self-confidence and in terms of the impact of digital stress (Figure 6). This observation contradicts the results of other studies that have analyzed the gender gap in digital competence in the Latin American and Caribbean areas, which generally benefits men with respect to their digital competence [62,63,64]. 

Older professors express having lower digital competence and lower ability to adapt (Figure 6), in general, to DLEs (Table 7). This observation is consistent with some previous works, which identify the older population as the most vulnerable in the digitization process [65]. If the psychological impact of the pandemic is analyzed at a general level, not necessarily in the digital domain, previous studies do not identify significant gaps by age [66], which shows the impact of the pandemic in the digital educational domain has specific behavior. In contrast, digital pandemic stress is, on average, lower among those over 55 years of age than the rest (Figure 6), which may be due to the greater maturity that is attributable to age and probably dampens the psychological impact of the pandemic among older professors, but this should be confirmed with an appropriate correlational study.

A strong gap was also found in areas of knowledge with respect to the variables studied (Table 8). In countries with low GII, professors from humanistic-social areas (Humanities and Social Sciences) are those who express greater digital competence, but in countries with high GII, it is the professors from scientific-technical areas (Science, Heath, and Engineering) who express more developed digital skills and greater self-confidence in the use of DLEs (except in Health Sciences, in the case of the variable self-confidence). On the other hand, professors in scientific-technical areas in countries with high GII are those who give greater importance to the technical and human resources linked to the use of DLEs. As for digital pandemic stress, among those in countries with low GII, those who express the highest level of stress are professors in Science and Health, while in countries with high GII, it is those in Humanities. 

Consequently, the impact of the pandemic on the adaptation to DLEs of professors in the different areas of knowledge depends strongly on the GII area of the country where the university is located. Specifically, as can be seen in the summary graph in Figure 7, in Humanities, Sciences and Health Sciences, the highest levels of pandemic stress are reached in countries with extreme values of their GII—low or high—in Social Sciences, stress increases when the GII grows and in Engineering the growth of the GII causes a decrease in the level of digital stress. As far as it has been possible to explore, there are no previous analogous studies with which to compare the results, so these conclusions constitute an element of originality of this work. There are studies that address the impact of the pandemic on publications on digital competence in teaching, which conclude that the highest frequency of publications is in Social Sciences [67]. 

### 4.2. Future Directions

The study did not use any exclusion criterion linked to the psychological state of the participants at the time of answering the survey. This fact may constitute a limitation of the study. As a future line of research, it would be useful to explore the academic or sociological reasons underlying the differences observed between GII zones, genders, and ages, in relation to the variables analyzed, carrying out a correlational study. It would also be interesting to identify the latent factors that, together with those studied, influence the determination of the pandemic stress studied. Finally, a comparative study of the perceptions of university professors from areas that differ in the GII would allow us to deepen the degree of influence of the GII in the development of stress due to the use of DLEs. 

Given that higher education is one of the sectors that has undergone the most intense digital migration process, professors are probably among the professionals who have had to make the greatest digitization effort due to the pandemic. Consequently, the results obtained on the digital pandemic stress of professors may be representative of the impact of the pandemic in other professional sectors, although more in-depth studies along these lines would be needed.

## 5. Conclusions

In the Latin American and Caribbean areas, university professors have an intermediate–high self-concept of their own digital competence and their ability to adapt to DLEs during the COVID-19 pandemic. However, the perception of the digital technical resources available to them and their self-confidence in the use of DLEs is lower. The level of stress caused by the use of DLEs during the pandemic is intermediate, but, according to the results of this study, it is not significantly conditioned either by the digital competence levels of the teachers or by the GII of the country of origin. This suggests that there are other sociodemographic, cultural, or political factors that also affect this type of stress and should be analyzed in future research.

Females generally report higher digital competence and lower digital pandemic stress than males. Older professors are those who express lower digital competence and also lower levels of digital pandemic stress, on average. In addition, there is a strong dependence of the indicated perceptions on the area of knowledge of the professors, the most affected by digital pandemic stress being those related to Health and Humanities.

## Figures and Tables

**Figure 1 behavsci-12-00203-f001:**
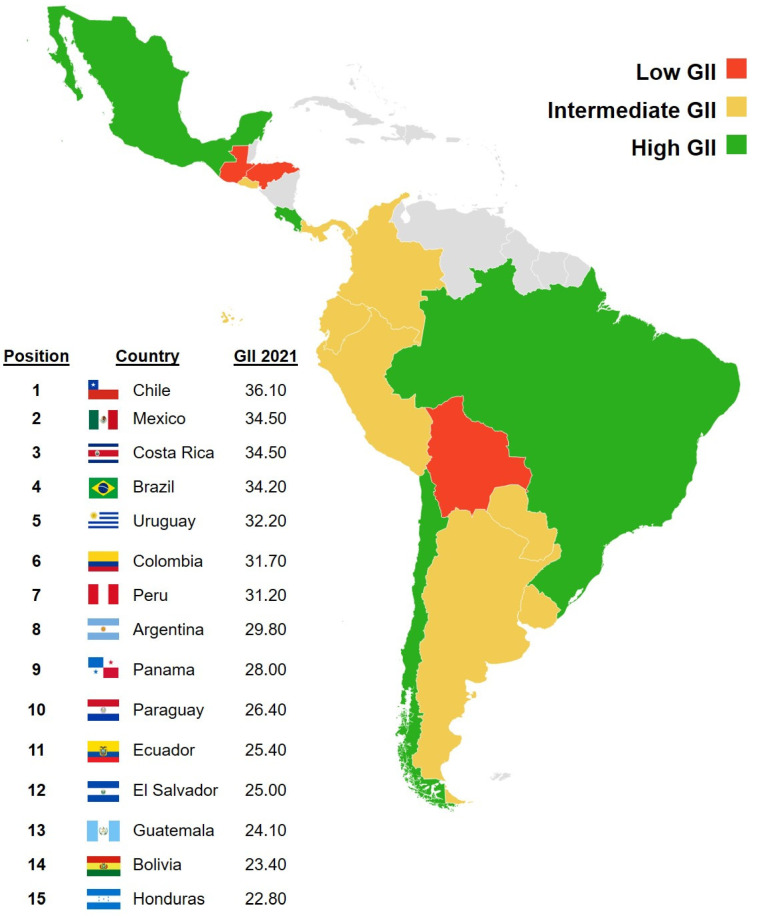
GII rank for Latin American and the Caribbean countries in 2021 whose GII is computed.

**Figure 2 behavsci-12-00203-f002:**
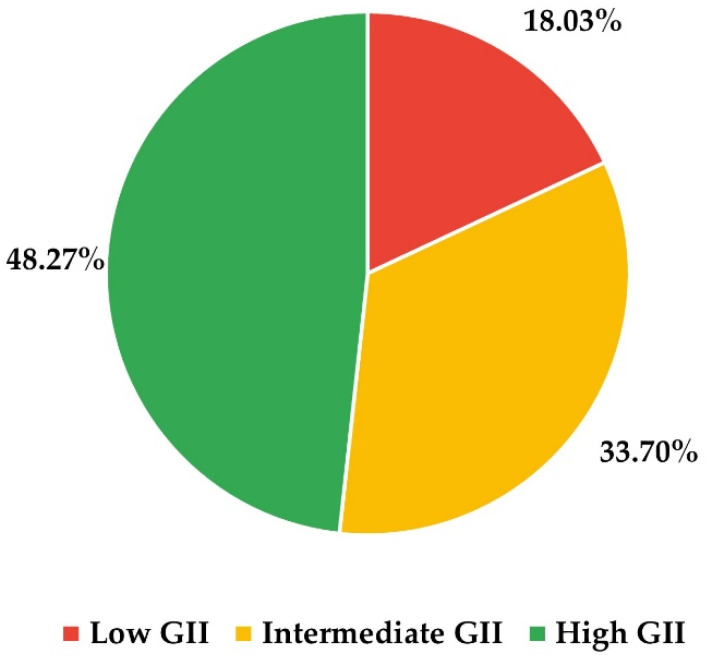
Distribution of the participants in the sample by GII area.

**Figure 3 behavsci-12-00203-f003:**
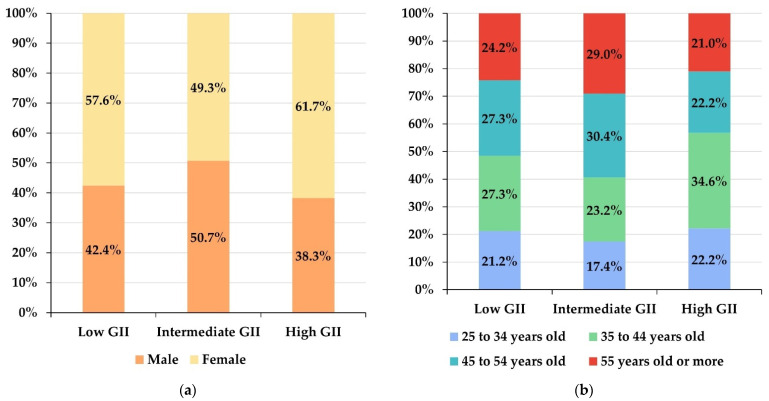
Distributions of the sample when the participants are differentiated by GII area and the rest of the independent variables: (**a**) by gender; (**b**) by age range; and (**c**) by area of knowledge.

**Figure 4 behavsci-12-00203-f004:**
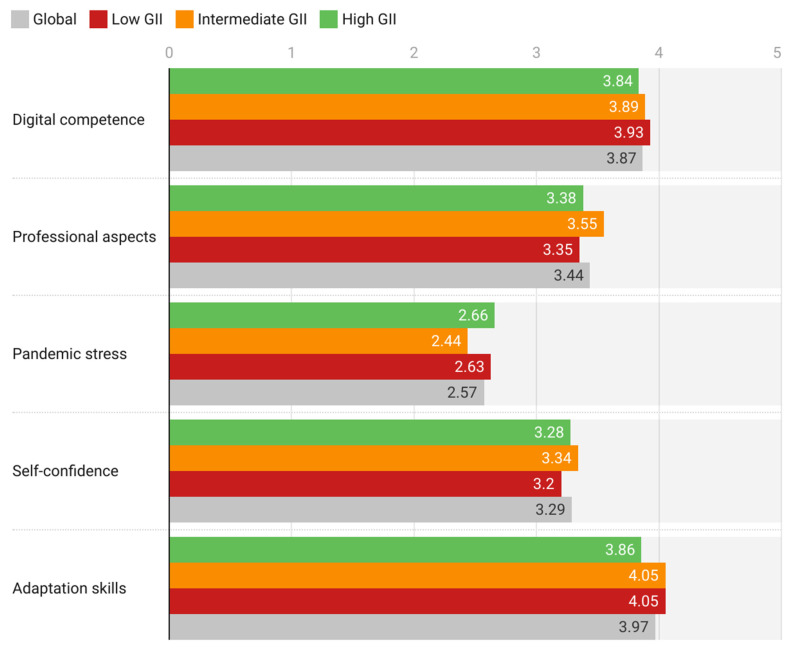
Mean values of the different subscales, globally and differentiated by GII area.

**Figure 5 behavsci-12-00203-f005:**
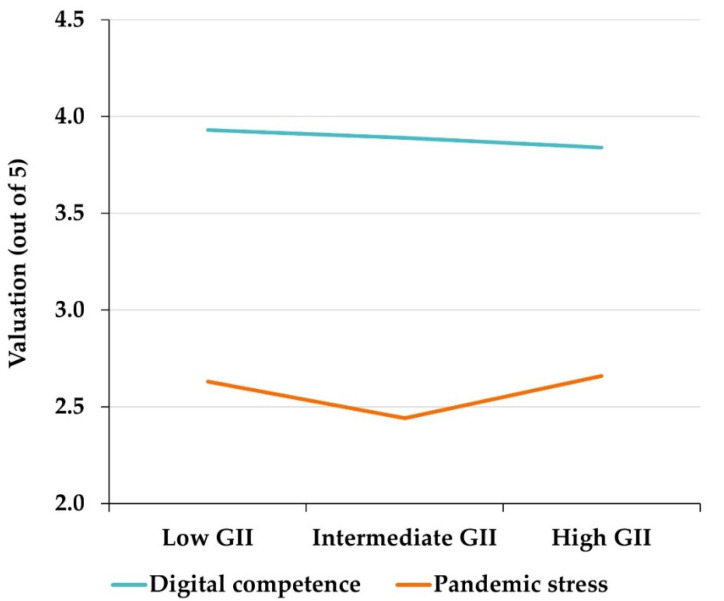
Variation in digital competence and pandemic stress from the use of DLEs, by country-of-origin GII.

**Figure 6 behavsci-12-00203-f006:**
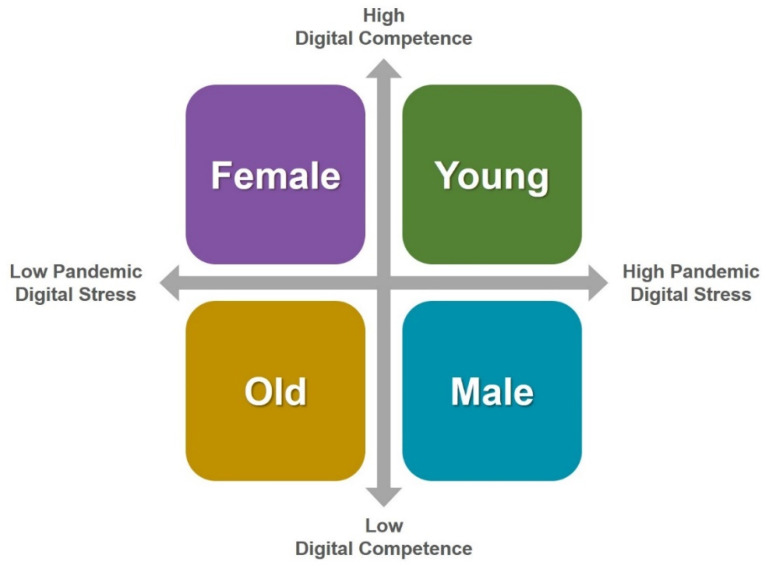
Grid of results on digital competence and digital pandemic stress by gender and age.

**Figure 7 behavsci-12-00203-f007:**
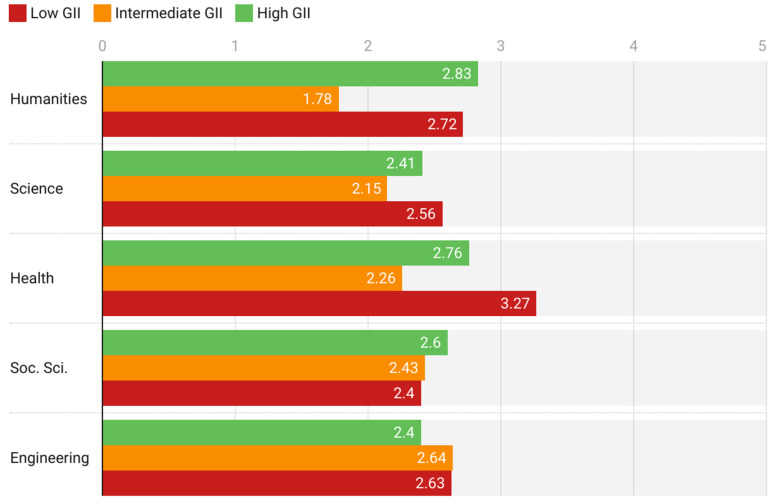
Variations in digital pandemic stress by GII in each area of knowledge.

**Table 1 behavsci-12-00203-t001:** Questions of the survey.

Variable	Item	Question
Assessment of own digital competence for the use of DLE during the pandemic (assessment of the indicated aspects)	Item 1	Agility
	Item 2	Continuous learning
	Item 3	Digital communication
	Item 4	Information management
	Item 5	Network leadership
	Item 6	Student orientation
	Item 7	Resilience
	Item 8	Teamwork
	Item 9	Strategic vision
	Item 10	Interest in increasing your digital skills in the future
	Item 11	Level of optimism with respect to the implementation of digital media for the teaching profession
Perception of the professional aspects linked to the use of DLE during the pandemic (assessment of the indicated aspects)	Item 12	University help
	Item 13	Faculty help
	Item 14	Student help
	Item 15	Spaces dedicated
	Item 16	Technical and human resources available to the professor for the use of DLE during the pandemic
Perception of pandemic digital stress	Item 17	Assess the tension you feel when using DLE
	Item 18	To what degree do you think the difficulties have increased in your work since the pandemic because of the need to use digital resources?
	Item 19	Are you upset about not being able to control certain aspects of your job related to DLE because of the pandemic?
	Item 20	Are you concerned that unexpected things may happen to you at work because of the pandemic?
	Item 21	Do you feel unable to cope with the digitization needs arising from the pandemic?
	Item 22	Assess the stress you feel due to the pandemic
	Item 23	Do you feel unable to control important aspects of your teaching activity because of the pandemic?
	Item 24	Assess anxiety about having to adapt to DLE
	Item 25	Do you feel distressed because of your job?
	Item 26	Assess the level of fear you feel towards COVID-19
	Item 27	Do you feel that you have less and less digital skills every day?
Self-confidence in the professor’s work during the pandemic	Item 28	I am currently feeling cheerful
	Item 29	Assess your emotional state during the pandemic on a scale of 1 to 5
	Item 30	Assess from 1 to 5 the level at which you have maintained an adequate relationship with your family and friends during the pandemic
	Item 31	I have been confident about my ability to handle my professional problems related to the pandemic
	Item 32	I have felt optimistic during the pandemic in relation to my teaching work
	Item 33	I have felt that I can control the academic difficulties that could appear due to the pandemic
	Item 34	I have felt that I have everything under control in relation to the development of the teaching activity during the pandemic
Self-concept on adaptation skills to DLE use during the pandemic	Item 35	I feel comfortable when teaching through DLE during the pandemic
	Item 36	Assess your adaptation as a professor to the use of DLE during the pandemic
	Item 37	Assess the training received on digital competence and the use of DLE during the pandemic
	Item 38	Do you consider developing objectives in the future to increase your digital competence in the wake of the pandemic?

**Table 2 behavsci-12-00203-t002:** Factorial weights and factors defined by the EFA.

Item	Factor 1 Digital Competence	Factor 2 Professional Aspects	Factor 3 Pandemic Stress	Factor 4 Self-Confidence	Factor 5 Adaptation Skills
Item 1	0.802				
Item 2	0.808				
Item 3	0.788				
Item 4	0.781				
Item 5	0.771				
Item 6	0.732				
Item 7	0.823				
Item 8	0.760				
Item 9	0.752				
Item 10	0.739				
Item 11	0.810				
Item 12		0.511			
Item 13		0.538			
Item 14		0.513			
Item 15		0.730			
Item 16		0.714			
Item 17			0.551		
Item 18			0.665		
Item 19			0.633		
Item 20			0.729		
Item 21			0.541		
Item 22			0.829		
Item 23			0.650		
Item 24			0.705		
Item 25			0.592		
Item 26			0.669		
Item 27			0.730		
Item 28				0.626	
Item 29				0.719	
Item 30				0.711	
Item 31				0.645	
Item 32				0.531	
Item 33				0.766	
Item 34				0.750	
Item 35					0.770
Item 36					0.674
Item 37					0.572
Item 38					0.638

**Table 3 behavsci-12-00203-t003:** Cumulative proportion of explained variance of the principal component analysis.

Item	Digital Competence	Professional Aspects	Pandemic Stress	Self-Confidence	Adaptation Skills
Proportion Variance	0.226	0.140	0.103	0.085	0.057
Cummulative Variance	0.226	0.366	0.469	0.554	0.611

**Table 4 behavsci-12-00203-t004:** Cronbach’s alphas, CR and AVE of the subscales.

Subscale	Cronbach’s Alpha	CR	AVE
Digital competence	0.89	0.88	0.65
Professional aspects	0.84	0.81	0.60
Pandemic stress	0.87	0.83	0.62
Self-confidence	0.82	0.80	0.60
Adaptation skills	0.77	0.74	0.54

**Table 5 behavsci-12-00203-t005:** Pearson correlation coefficients of the different subscales among themselves and with respect to the global survey.

	Digital Competence	Professional Aspects	Pandemic Stress	Self-Confidence	Adaptation Skills	Global
Digital competence	1	0.3021	−0.0531	0.3786	−0.0688	0.8219
Professional aspects		1	−0.0107	0.3054	−0.0095	0.7622
Pandemic stress			1	0.1203	−0.1169	0.7011
Self-confidence				1	−0.0617	0.7740
Adaptation skills					1	−0.7097
Global						1

**Table 6 behavsci-12-00203-t006:** Mean values differentiated by gender and MANOVA test statistics.

Subscale	Male			Female			MANOVA’s F	MANOVA’s *p*-Value
	Low	Intermediate	High	Low	Intermediate	High		
Digital competence	3.74	3.81	3.80	4.07	3.97	3.87	9.9889	0.0000 *
Professional aspects	3.17	3.42	3.41	3.48	3.69	3.36	9.5868	0.0000 *
Pandemic stress	2.79	2.35	2.65	2.51	2.53	2.66	16.8986	0.0000 *
Self-confidence	3.16	3.49	3.27	3.22	3.18	3.29	9.4527	0.0001 *
Adaptation skills	3.80	3.92	3.84	4.22	4.18	3.88	8.8122	0.0002 *

* *p* < 0.05.

**Table 7 behavsci-12-00203-t007:** Mean values differentiated by age and MANOVA test statistics.

Subscale	<35 Years Old			35–44 Years Old			45–54 Years Old			≥55 Years Old			F	*p*-Value
	L.	I.	H.	L.	I.	H.	L.	I.	H.	L.	I.	H.		
Digital competence	4.53	4.17	4.06	4.17	3.89	3.95	4.26	3.65	4.00	3.63	3.97	3.27	88.0046	0.0000 *
Professional aspects	3.14	3.47	3.36	3.56	3.69	3.42	3.33	3.39	3.60	3.33	3.66	3.11	8.3495	0.0000 *
Pandemic stress	2.56	2.73	2.84	2.80	2.50	2.78	2.64	2.36	2.44	2.49	2.30	2.49	4.6898	0.0000 *
Self-confidence	2.79	3.48	2.82	3.06	3.28	3.46	3.47	3.13	3.38	3.41	3.51	3.37	14.1711	0.0000 *
Adaptation skills	4.04	4.21	4.04	4.08	3.86	3.88	4.25	3.99	3.97	3.78	4.16	3.53	10.5530	0.0000 *

* *p* < 0.05.

**Table 8 behavsci-12-00203-t008:** Mean values differentiated by area of knowledge and MANOVA test statistics.

Subscale	Humanities			Science			Health			Soc. Sci.			Engineering			F	*p*-Value
	L.	I.	H.	L.	I.	H.	L.	I.	H.	L.	I.	H.	L.	I.	H.		
Digital competence	4.36	4.02	4.08	3.69	3.86	4.00	3.42	4.00	3.99	4.58	3.74	3.72	3.55	3.94	3.61	49.2770	0.0000 *
Professional aspects	3.56	3.80	3.60	3.24	3.55	3.10	2.93	4.04	3.57	3.77	3.31	3.32	3.14	3.53	3.26	9.3167	0.0000 *
Pandemic stress	2.72	1.78	2.83	2.56	2.15	2.41	3.27	2.26	2.76	2.40	2.43	2.60	2.63	2.64	2.61	20.4670	0.0000 *
Self-confidence	3.45	3.25	3.20	2.83	3.19	2.94	3.42	3.67	3.04	3.46	3.31	3.42	3.18	3.29	3.41	4.8637	0.0000 *
Adaptation skills	4.35	4.35	4.03	3.68	4.00	4.16	4.42	4.22	4.00	4.61	3.81	3.54	4.11	4.12	3.90	20.3680	0.0000 *

* *p* < 0.05.

## Data Availability

The data are not publicly available because they are part of a larger project involving more researchers. If you have any questions, please ask the contact author.

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
