# Peer review of "Influence of Country Digitization Level on Digital Pandemic Stress"

_behavsci, 2022, doi:10.3390/bs12070203_

Round 1
Reviewer 1 Report
This is a very good paper investigating the influence of the COVID-19 pandemic on the adaptation of university professors to digital learning environments (DLE). The paper is well-written and of interest for the readers, particularly those interested and working n education practices. I would recommend minor changes before considering the paper for publication.
In the introduction section, the authors are reporting the main changes in the technical resources, for instance, the virtual communication platforms. I would also add a paragraph summarizing which techniques or platforms were existing before the COVID-19 pandemic and were not usually used. The aim of achieving digitalization was necessary before the pandemic.
The last paragraph of the introduction section reports the main aim of the study as "the main interest of the paper". I would prefer to report it as "the main goals were to analyze...".
Was previous psychological or psychiatric diagnosis an exclusion criteria in the study?
Subsection 2.4 is called "Procedures". I would recommend to rename it as "Statistical analyses". Procedures should be considered those variables and assessments that were considered.
Subsection 3.1., from the results section, is called "Analysis of participants". This is not a good title for the main description of the sample. I would prefer to rename it as "Main characteristics of the total sample".
Lines 198-200. I would recommend to rephrase it. What does it mean: By gender, women are... It is not necessary to mention "By gender".
Subsection 3.2, (line 207): The title of the section should be renamed. Analysis of the instrument is apparently refering to main description of the survey. Psychometric validation was a first step. It should be reflected in the title of the subsection.
The participants expressed intermediate-high levels of digital competence. Was then possible that the teaching process for the digitalization for professors was not adequately made?
The stress caused by the COVID-19 pandemic had an influence on aspects of the professional activity, for instance, on research activity. Which kind of interventions should be applied when taking into account the necessary activity in research?
The last paragraph of the discussion section, is mainly focused on the future lines of research of the topic. I consider very important this section, so I recommend to state it as a separate subsection called "Future directions".
Author Response
Please, find enclosed a detailed response.

Reviewer 2 Report
1. Add information about the population in the title i.e. professors
2. Provide information about the state of the art of the study and analyze the gap of the study in the introduction.
3. Are they the aims of the study? If yes, they are several differences such as to identify significant gaps by gender, age or 126 area of knowledge in the analyzed perceptions among professors from each GII area. It is not mentioned in the final part of the introduction.
4. If the aim is identify significant gaps by gender, age or 126 area of knowledge in the analyzed perceptions among professors from each GII area, then the authors have to provide the assumptions of significances of each characteristic in the introduction.
5. Do the items in the instrument developed from a specific theory, model, or framework? If yes, please explain it. If not, it can be a limitation in this study. Please explain its validity and reliability as well.
Author Response

(The authors gave the same response as above.)

Reviewer 3 Report
Thank you very much for the opportunity to read your paper.
I found no errors in terms of methodology and interpretation of the results. The text is apparent and understandable.
I have only two small recommendations:
1. it would be helpful to provide a literature search strategy to conceptualize the theoretical background and discussion. This is a topic that is already very widely developed, and the way you search the literature may influence your interpretations.
2. I feel that Figures 5, 6, and 7 should be part of the results rather than the discussion.
Author Response

(The authors gave the same response as above.)

Reviewer 4 Report
The paper meets Journal areas. The paper's aims are actual and valuable under the ongoing consequences of COVID-19. However, the paper should be improved before publishing.
Introduction: It should be better to add the structure of the paper at the end of the Introduction. Besides, the introduction should briefly justify the actuality of the investigation.
It would be better to add the literature review of the investigation on mentioned above issues. This part is missed in the paper at all.
Methods Add the hypotheses of the investigation.
Conclusion&Discussion. It would be better to add the comparison analysis of the research findings with the previous investigations. Hope it would be added, after adding the literature review.
Author Response

(The authors gave the same response as above.)

Round 2
Reviewer 4 Report
The authors considered my suggestions.